# Effects of *Spartina alterniflora* Invasion on Soil Carbon, Nitrogen and Phosphorus in Yancheng Coastal Wetlands

**Yufeng Sheng** [1], **Zhaoqing Luan** [1,*], **Dandan Yan** [2], **Jingtai Li** [1], **Siying Xie** [1], **Yao Liu** [1], **Li Chen** [1], **Min Li** [1] and **Cuiling Wu** [1]

1 College of Biology and the Environment, Co-Innovation Center for Sustainable Forestry in Southern China, Nanjing Forestry University, Nanjing 210037, China

2 College of Horticulture and Plant Protection, Henan University of Science and Technology, Luoyang 471023, China

* Correspondence: luanzhaoqing@njfu.edu.cn; Tel.: +86-136-7516-1587

**Abstract:** The rapid invasion of *Spartina alterniflora* threatens the ability of soils to store carbon (C), nitrogen (N) and phosphorus (P) in coastal wetlands. This study analyzed the temporal and spatial distribution characteristics of soil C, N and P in *Spartina alterniflora* wetland in the Dafeng Elk Nature Reserve of Yancheng, China, in different invasion stages from 1995 to 2020. The results suggest that: (1) the invasion of *Spartina alterniflora* increased the content and storage of soil C and N, and decreased the content and storage of soil P; (2) altered soil properties caused by *Spartina alterniflora* invasion also indirectly affected the accumulation of soil C, N and P in wetland ecosystem. Organic carbon (SOC), total nitrogen (TN) and total phosphorus (TP) were positively correlated with soil moisture content (SMC) and electrical conductivity (SEC), and negatively correlated with bulk density (SBD) and pH; (3) *Spartina alterniflora* invasion increased soil C/P and N/P, and decreased soil C/N. In conclusion, *Spartina alterniflora* invasion has changed the ecosystem, increased the storage capacity of soil C and N in the invasive ecosystem, but reduced the storage capacity of soil P to a certain extent.

**Keywords:** *Spartina alterniflora* invasion; soil carbon; soil nitrogen; soil phosphorus; ecological stoichiometry

## 1. Introduction

The coastal wetland ecosystem is in the transition zone between land and sea [1] and is one of the most diverse and valuable ecosystems in the world [2–4]. It has unique ecological processes that combat coastal erosion, regulate the climate [5], purify water and provide habitats for organisms. Coastal wetlands have a high carbon (C) sequestration capacity and are major contributors to China's blue carbon sink [6,7], as well as being important to the global biogeochemical cycle [3,8]. However, coastal wetlands are vulnerable to the threat of species invasion due to the fragility of their ecosystems. Biological invasions are an important issue in global change. Invasion by exotic plants significantly alters local ecosystems [9]; it can alter local soil nutrient pools and elemental cycles [10–14], change the structure and function of the soil microbiome [15,16] and also have an impact on the abundance, diversity and structure of local plant and animal communities [17–19]. Our country introduced *Spartina alterniflora* in the 1980s [20] and it has served to protect shorelands. However, due to its broad salinity [21,22], strong flooding tolerance and high reproduction coefficient [23], *Spartina alterniflora* has expanded its habitat area to coastal wetlands in China at an exponential growth rate, replacing native plants as the dominant species in a short period of time [24], affecting biochemical pools and energy flow [25] and altering the basic structure and function of native ecosystems.

The biogeochemical cycles of C, N and P have a significant impact on global environmental change and have attracted the attention of scholars worldwide. It has been demonstrated that the invasion of coastal wetlands by *Spartina alterniflora* can have a great

impact on the soil Cand N pools of wetlands [26–29], which can greatly increase the content of soil organic carbon (SOC) and total nitrogen (TN) [30,31], thus affecting the processes and functions of the whole ecosystem. Understanding the changes in soil N and P content following *Spartina alterniflora* invasion is valuable for understanding the invasion mechanism of *Spartina alterniflora*. Ecological stoichiometry mainly refers to the stoichiometric relationship between C, N and P. C/N and C/P can be good indicators of the carbon enrichment capacity of soils. Ecological stoichiometry studies can reveal the theory of chemical transformation of nutrients and nutrient regulation mechanisms among ecosystem components, helping us to understand the nutrient mechanisms of invasive species and the important factors driving dynamic nutrient balance in the environment [32–34]. Considering the different impacts of *Spartina alterniflora* upon invasion of different coastal wetlands, more field studies are necessary to fully understand the alteration of soil nutrient dynamics by *Spartina alterniflora* invasion.

As is well-known, the invasion of *Spartina alterniflora* largely affects the survival and growth of native species. Compared to many native species, *Spartina alterniflora* has greater biomass and a more developed root system, and the low rate of soil respiration in *Spartina alterniflora* communities slows the decomposition of SOC, making *Spartina alterniflora* communities highly capable of carbon sequestration [35,36]. Similarly, *Spartina alterniflora* invasion significantly increases the N content of wetland soils, and excess N has the potential to further enhance *Spartina alterniflora* invasion [37,38]. In addition, increased soil N content also significantly affects *Spartina alterniflora* growth rates and biomass allocation. P cycling is also strongly influenced by *Spartina alterniflora* invasion [39], not only in terms of N content, but also in the transformation between N forms [22,40], and it has been suggested that the N content of *Spartina alterniflora* communities also tends to decrease along the landward direction and with increasing soil depth [41]. However, many studies have also shown that the invasion of H. *Spartina alterniflora* has no significant effect on soil properties, biomass accumulation, or the C, N and P storage capacity of plants or soils in the invaded sites, and may even lead to a decrease in C, N and P storage in the plant–soil system [42]. For example, the invasion of mangrove communities by *Spartina alterniflora* resulted in a decline in their soil C sinks [43,44]. Other studies have shown that soil C, N and P contents and stocks vary at different times of invasion. The effects on soil C, N and P differ depending on the stage of *Spartina alterniflora* invasion. Zhang et al. showed that organic carbon content began to decrease with increasing soil depth after seven years of *Spartina alterniflora* invasion [45], Xu et al. also found that SOC content increased and then decreased only on a decadal scale after *Spartina alterniflora* invasion, rather than increasing monotonically [43], and Xie et al. showed that *Spartina alterniflora* increased sediment nutrient levels early in the invasion and then significantly decreased total C, TN and TP [46].

Although there have been some scholars who have studied the physicochemical properties of coastal wetland soils after the invasion of *Spartina alterniflora*, most of them have been limited to the content of a certain chemical element and have not combined the interaction between multiple chemical elements. In addition, most previous investigations have focused on shallow soil layers, which may have led to uncertainty in the results of the studies, so there is a need to combine the results of studies from soil profiles to 100 cm depth. There were also large differences in the response of soil C, N and P to the timing of *Spartina alterniflora* invasion, and current studies lack a comprehensive understanding of the effects of *Spartina alterniflora* invasion time series on soil SOC and N and P dynamics.

Therefore, in order to assess the ecosystem response to invasion by alien plants and to better understand the spatial and temporal distribution of soil C, N and P in coastal wetlands, we investigated the soil C, N and P contents, stocks and stoichiometric ratios in coastal wetlands in Yancheng Dafeng Elk Nature Reserve after invasion by *Spartina alterniflora*, with the aim of providing a theoretical basis for wetland biological element cycling, wetland management and conservation activities.

## 2. Materials and Methods

### 2.1. Study Area

Dafeng Elk Nature Reserve is located south of the Yellow Sea wetlands in Jiangsu Province, one of the four largest wetlands in China. The land is flat and consists of mostly tidal saline and early saline soils. With a total area of 25 km$^2$, the reserve is the largest wild elk nature reserve in the world. The climate of Dafeng Elk Nature Reserve is mainly a warm, temperate monsoon continental climate, with an annual average temperature of about 14.1 °C and significant marine monsoon characteristics. The average annual precipitation in the reserve is 1068 mm, with 116.4 days of precipitation per year [47]. In this study, the third core area of the Dafeng Elk Nature Reserve was selected as the study area (Figure 1), which is densely vegetated, mainly with *Spartina alterniflora* rice grass, alkali ponies and reeds. Since its introduction to the reserve in 1993, *Spartina alterniflora* rice grass has replaced reed and alkali poncho as the dominant species in the study area [48].

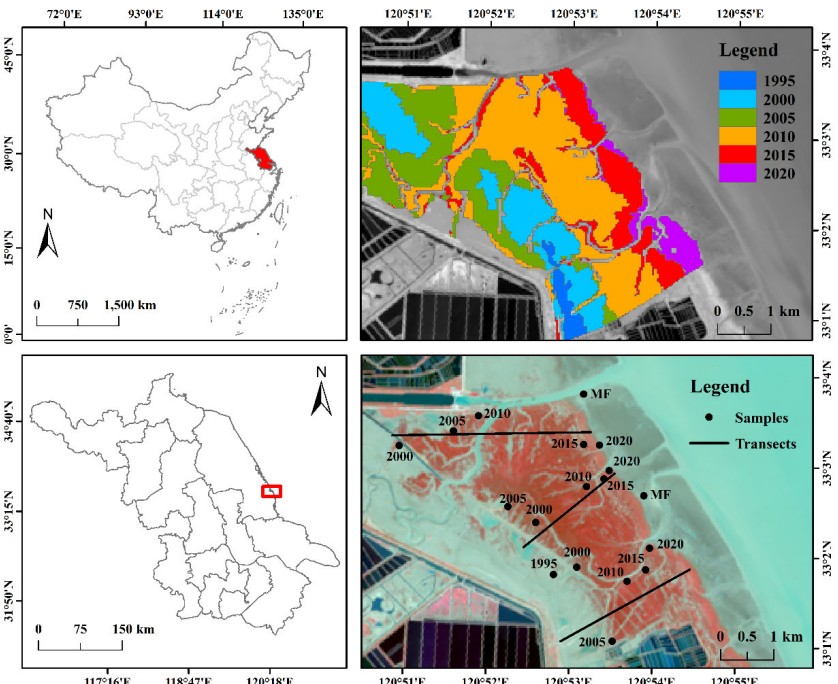

**Figure 1.** Location and sampling site map.

### 2.2. Sample Collection and Laboratory Analysis

(1) In this study, the *Spartina alterniflora* community was divided according to different invasion stages. Yan et al. combined GEE and object-based hierarchical RF classification to obtain spatial data on the invasion dynamics of *Spartina alterniflora* milfoil in the third core area of Dafeng Elk Nature Reserve from 1993 to 2020 [48]. Based on these data, the *Spartina alterniflora* communities and light beaches growing in 1995, 2000, 2005, 2010, 2015 and 2020 were selected for this study at five-year intervals (denoted as SA26, SA21, SA16, SA11, SA6, SA1 and MF, respectively). Among them, MF was used as the control for the invasion of *Spartina alterniflora*. A total of three sample lines with 50 sampling points were set up according to the topography and the growth trend of *Spartina alterniflora*. The geographical locations of the sampling points were determined by overlaying the spatial data of *Spartina alterniflora* with remote sensing images.

(2) In November 2020, plant and soil samples were taken. A total of three sets of replicates were set up for each sampling site. Soil samples were also collected using an original soil auger at a depth of 1 m (4.5 cm diameter) and stratified according to the set depth (20 cm for one layer, five layers in total.). Soil samples were manually removed from plant roots, shells, gravel and other debris and then naturally dried, milled, sieved

and sealed. The aboveground parts were cut from the sample plots after measuring and recording the height and cover of the *Spartina alterniflora*, brought back to the laboratory and dried at 85 °C for 48 h to determine the dry weight of the plants.

(3) TC and TN were determined using an elemental analyzer (vario MACRO cube, Germany) in CNS mode. Determination of soil TP was done using the molybdenum–antimony colorimetric method; determination of soil SOC content was done using the standard wet oxidation method (Walkley–Black technique). Soil bulk density (SBD) and soil moisture content (SMC) were calculated using the drying method [49]. Soil pH (pH) and soil electrical conductivity (SEC) were measured using a pH meter and conductivity meter, respectively.

*2.3. Data Analysis*

One-way ANOVA was used to compare differences in soil physicochemical properties, SOC, TN and TP at different years of invasion and at different depths of *Spartina alterniflora*. Linear relationships between soil physicochemical properties and soil SOC, TN and TP were analyzed using Pearson correlations. For all tests, differences were considered significant if $p < 0.05$. The data in the text were analyzed using SPSS v22.2 and graphs were plotted and correlated using OriginPro 2021b.

## 3. Results

*3.1. Changes in Vegetation Characteristics and Soil Physicochemical Properties of Spartina alterniflora at Different Years of Invasion*

The biomass and plant height of *Spartina alterniflora* differed between invasion stages, with a general trend of increasing and then decreasing, the maximum biomass being recorded at 11 years of invasion. The number of plants of *Spartina alterniflora* tended to fluctuate upwards. (Table 1).

**Table 1.** Characteristics of plants of different invasive years.

| Sample | Aboveground Biomass (AGB, g m$^{-2}$) | Plant Height (cm) | Plant Density (n m$^{-2}$) |
|---|---|---|---|
| SA1 | 1816.92 ± 373.78 b | 157.71 ± 11.70 b | 394.44 ± 65.88 b |
| SA6 | 2321.08 ± 350.51 b | 169.71 ± 13.00 b | 354.17 ± 81.27 b |
| SA11 | 3178.08 ± 527.66 a | 194.78 ± 14.64 a | 338.89 ± 76.15 b |
| SA16 | 2339.5 ± 591.03 b | 134.33 ± 4.72 c | 587.50 ± 72.02 a |
| SA21 | 1737.56 ± 335.84 b | 117.56 ± 9.61 d | 535.71 ± 65.92 a |
| SA26 | 541.67 ± 149.31 c | 44.00 ± 3.61 e | 625.00 ± 139.36 a |

Note: The lowercase letters after each column of figures represent significant differences in different years of invasion ($p < 0.05$).

From Figure 2, it can be seen that there are significant differences in soil physicochemical properties between the different invasion stages of *Spartina alterniflora* in Dafeng Elk Reserve. The invasion of *Spartina alterniflora* increased SMC and decreased pH, SBD and SEC. Horizontally, SMC showed the opposite trend to pH and SBD, increasing in the early stage of invasion and then decreasing. In the vertical direction, soil pH of *Spartina alterniflora* communities did not differ significantly at different soil depths during the same invasion stage. Except for the SA21 and SA26 samples, the SBD (0–20 cm) was less than that of the underlying soil, with the maximum value occurring in the 20–40 cm soil layer. SMC did not differ significantly between soil depths for all invasion stages of *Spartina alterniflora* except for the SA11 and SA16 samples; SEC did not differ significantly between soil depths for all invasion stages except for the SA11 sample.

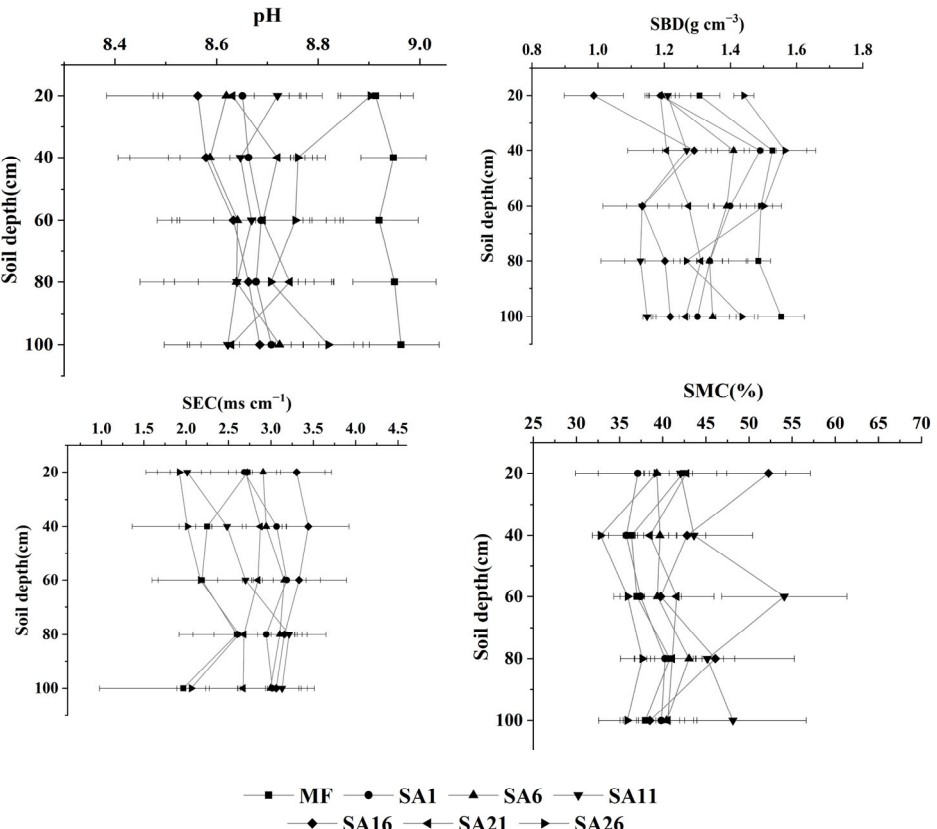

**Figure 2.** Deep changes in soil properties at different stages of invasion.

### 3.2. Changes in SOC at Different Years of Invasion

The SOC content of the mudflat soils did not differ significantly between the different soil layers (Figure 3). Soil SOC accumulated after the invasion of *Spartina alterniflora*, and SOC content tended to increase and then decrease with the number of years of invasion. Compared with the MF, the average SOC content of the SA1, SA6, SA11, SA16, SA21 and SA26 samples increased by 31.4%, 79.5%, 114.3%, 147.9%, 107% and 110.9% respectively. Soil SOC content reached a maximum in the 16th year of *Spartina alterniflora* invasion and is about 2.5 times higher than the SOC content of the light beach soils. The greatest variation in SOC content was found in the 0–20 cm soil layer, where SA16 had the greatest SOC content, with a 419.5% increase over MF. The deep organic carbon content of SA26 soils was significantly higher than in the other sample sites.

Soil SOC stocks at all soil depths were much smaller in the mudflats than in the *Spartina alterniflora* sample (Figure 3). Soil SOC stocks increased with the number of years of invasion of *Spartina alterniflora* between 1 and 16 years of invasion but declined and then increased between 16 and 21 years of invasion. Compared to SA16, the soil in the SA21 sample site had 19% less SOC storage in the 0–100 cm soil layer. In terms of vertical distribution, the maximum soil SOC stocks in SA1 and SA6 were found at 40–60 cm, the maximum soil SOC stocks in SA11 were found at 20–40 cm, the maximum soil SOC stocks in SA16 and SA21 were found at 0–20 cm on the soil surface, while the soil SOC in SA26 was evenly distributed at all soil depths with no significant differences.

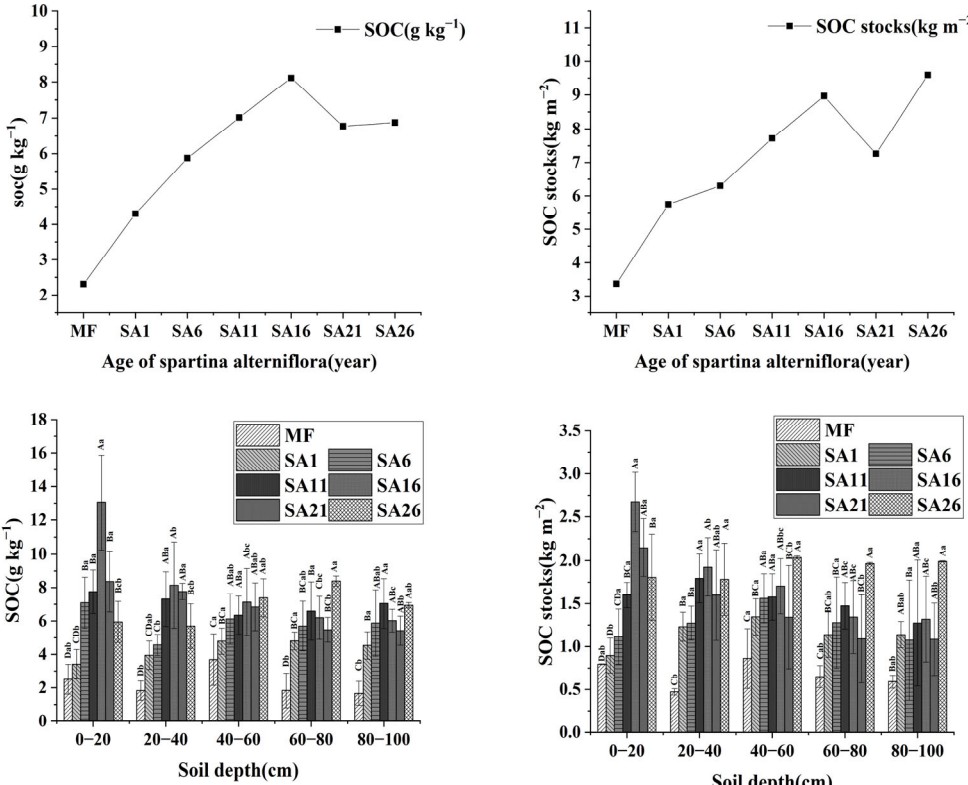

**Figure 3.** Distribution characteristics of soil SOC content and storage at different stages of invasion. Different letters indicate significant differences between different years of invasion or soil depths. (*p* < 0.05).

### 3.3. Changes in TN and TP at Different Years of Invasion

The TN content of the mudflat soil was significantly lower than that of the *Spartina alterniflora* samples at all stages of invasion (Figure 4). In line with the trend in SOC content, the soil TN content of the *Spartina alterniflora* samples also tended to increase and then decrease with increasing age of invasion, also peaking at 16 years of invasion (about 2.9 times that of the mudflats). Compared with MF, the mean TN content of SA1, SA6, SA11, SA16, SA21 and SA26 increased by 57.1%, 104.3%, 157.1%, 186.2%, 146.1% and 109.1%, respectively. The soil TN content of the SA16 sample was significantly greater than the other *Spartina alterniflora* samples at 0–60 cm soil depth, but at 60–100 cm depth, the SA11 sample had much greater soil TN content than the other samples. TN content in the SA26 *Spartina alterniflora* sample was higher in the deeper soil layers (60–100 cm) than in the newer *Spartina alterniflora* sample.

Soil TN stocks were significantly greater in the *Spartina alterniflora* sample than in the bare beach soils (Figure 4). *Spartina alterniflora* significantly increased soil TN stocks in the early stages of invasion (1–6 years), and after 6 years of invasion, the increase in soil TN stocks slowed. In contrast, after 21 years of invasion, soil TN stocks declined, with the SA26 sample having 15% fewer TN stocks in the 0–100 cm soil layer than the SA21 sample. Soil TN stocks in the SA1 sample reached a maximum at 40–60 cm, and soil TN stocks in the SA6 sample reached a maximum at 20–40 cm. The maximum values of soil TN storage in the SA11, SA16 and SA21 samples all occurred at 0–20 cm. Soil TN stocks in the SA26 sample site were evenly distributed at all depths with no significant differences.

Soil phosphorus content did not respond significantly to the invasion of *Spartina alterniflora*. The TP content of the soil increased when *Spartina alterniflora* invaded for one year (Figure 5), but decreased when *Spartina alterniflora* continued to invade, and then showed a trend of increasing and then decreasing as the number of years of invasion increased. The maximum value of soil TP content during the 26 years of invasion of

*Spartina alterniflora* occurred at 1 year of invasion and the minimum value at 26 years of invasion. Compared with MF, the TP content of SA1 increased by 2.5% and that of SA26 in the 0–100 cm soil layer decreased by 15.3%. Overall, the invasion of *Spartina alterniflora* reduced soil TP levels. In terms of vertical distribution, there were no significant differences in soil TP content at different soil depths for the same invasion stage.

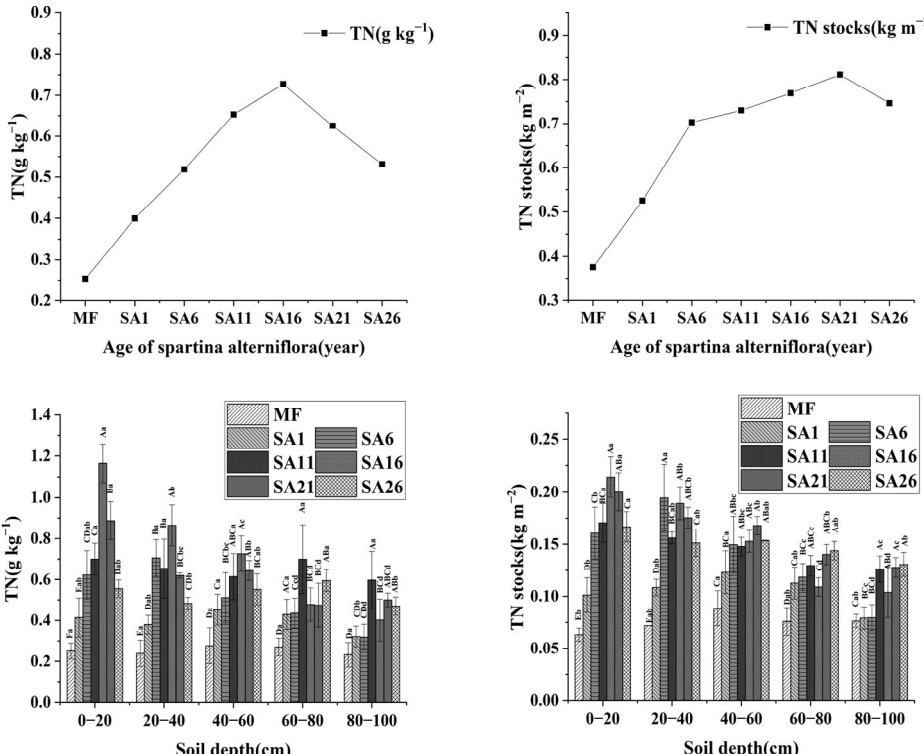

**Figure 4.** Distribution characteristics of soil TN content and storage at different stages of invasion. Different letters indicate significant differences between different years of invasion or soil depths. ($p < 0.05$).

Soil TP stocks and TP content in the *Spartina alterniflora* communities showed roughly the same trend (Figure 5). Similarly, soil TP stocks increased at the beginning of the invasion and then decreased, then increased and then decreased again with the number of years of invasion. In terms of vertical distribution, the maximum soil TP storage in samples SA1 and SA6 occurred at 20–40 cm, the maximum soil TP storage in sample SA16 occurred at the deeper layer (80–100 cm), the maximum soil TP storage in samples SA21 and SA26 occurred at 0–20 cm, while soil P stocks in the SA11 sample site were evenly distributed across the soil layers.

### 3.4. Ecological Stoichiometric Characteristics of C, N and P in Soils of Different Invasion Years

The mean C/N ratio of the bare beach soil was 43.880 and the mean C/N ratio of the *Spartina alterniflora* soil was in the range of 24–33, with the C/N ratio of the MF soil being significantly higher than that of the *Spartina alterniflora* sample (Figure 6). The lowest value of 24.248 was reached at 11 years of invasion, and the soil C/N ratio tended to increase between 11 and 26 years of invasion but was not significantly different. The soil C/P ratio and soil N/P ratio showed the same trend, and the mean values of the soil C/P ratio and N/P ratio in MF were 15.913 and 0.377, respectively, which were significantly smaller than the C/P ratio and N/P ratio of the *Spartina alterniflora* soil sample in the *Spartina alterniflora* community. The C/P and N/P ratios of the *Spartina alterniflora* sample showed a trend of increasing and then decreasing with increasing age of invasion, with the maximum value occurring at 11 years of invasion. Significant differences in C/P ratios and N/P ratios

were found in the *Spartina alterniflora* samples at all stages of invasion, except for the SA26 sample.

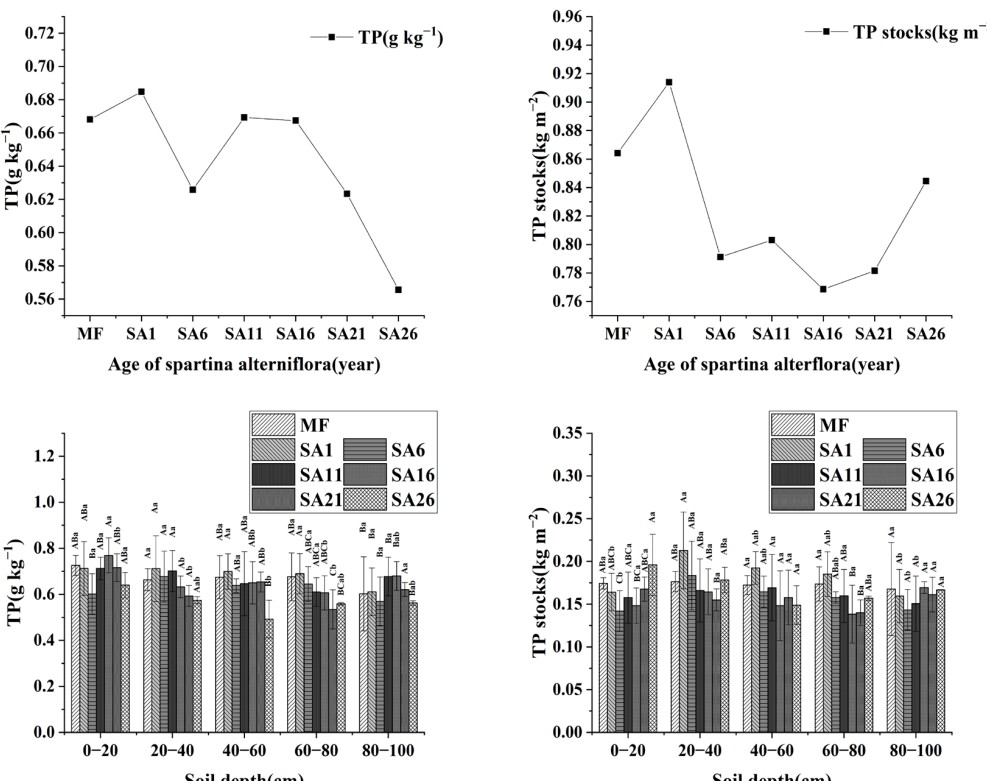

**Figure 5.** Distribution characteristics of soil TP content and storage at different stages of invasion. Different letters indicate significant differences between different years of invasion or soil depths. ($p < 0.05$).

In terms of vertical distribution, C/N ratios did not vary significantly from 0–100 cm of soil in each *Spartina alterniflora* sample except in SA1 and SA6 samples, where C/N ratios were significantly higher at 80–100 cm than at 0–60 cm in SA1, and at 60–100 cm than at 0–40 cm in SA6. The C/P and N/P ratios of the *Spartina alterniflora* sample plots invaded for 1–21 years showed a trend of increasing and then decreasing with increasing age of invasion at 0–60 cm soil depth, and there were no significant differences between depths.

### 3.5. Correlation of Soil Properties with Soil C, N and P Contents and Their Biogenic Stoichiometric Characteristics

As shown in Figure 7, TC and TN are significantly correlated. Both TC and TN were positively correlated with SMC and SEC, significantly negatively correlated with SBD and pH, and TP was significantly negatively correlated with SBD and pH. C/N was significantly and positively correlated with SBD and negatively correlated with SMC, and both the C/P ratio and N/P ratio were significantly and positively correlated with SMC and significantly and negatively correlated with SBD. In addition, due to the small change in TP content and the extremely significant positive correlation between TC and TN, C/P and N/P are extremely significantly correlated. The aboveground biomass of *Spartina alterniflora* was significantly correlated with TC, TN, TP, C/P and N/P.

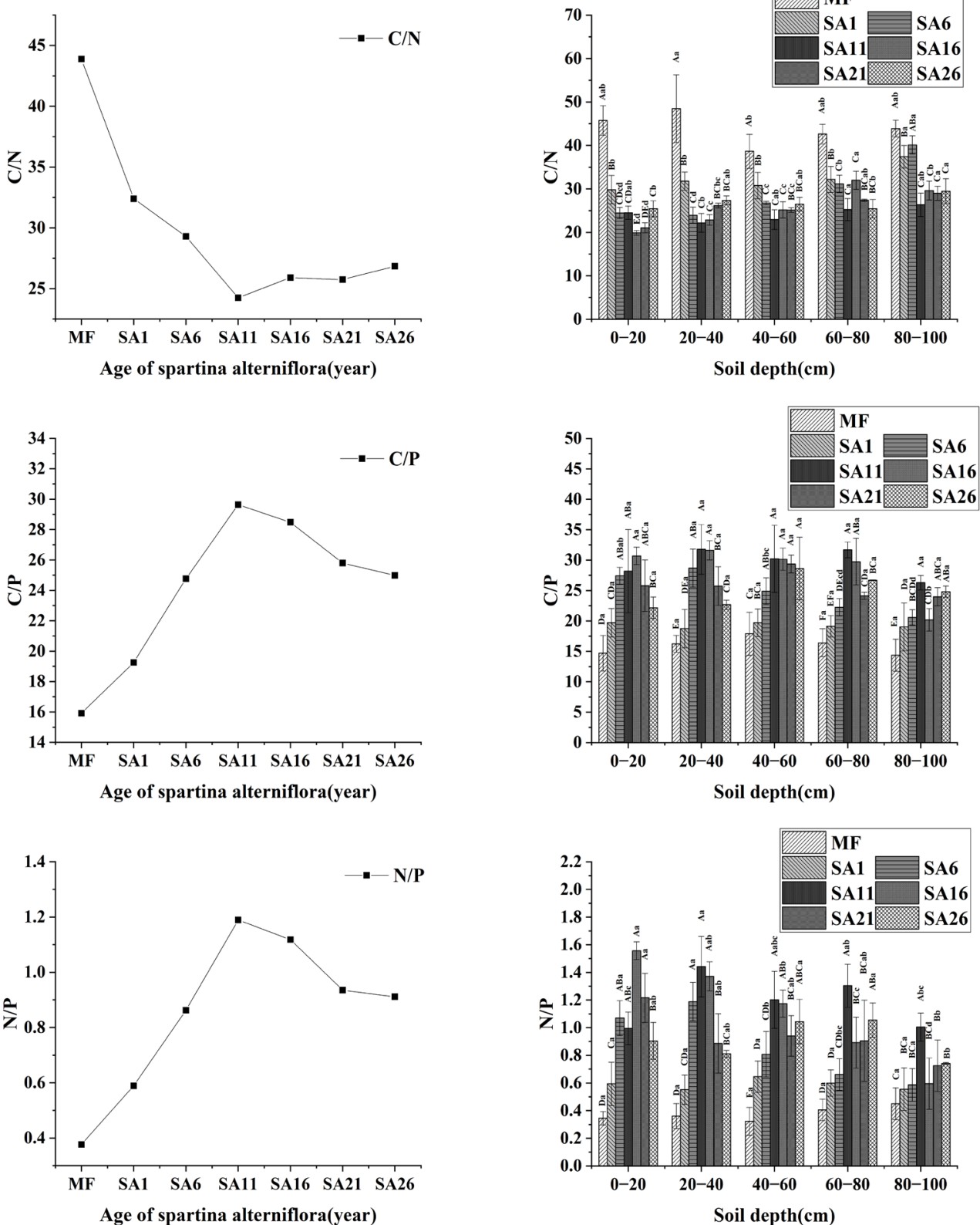

**Figure 6.** Ecological stoichiometry of soil C, N and P at different stages of invasion. Different letters indicate significant differences between different years of invasion or soil depths. ($p < 0.05$).

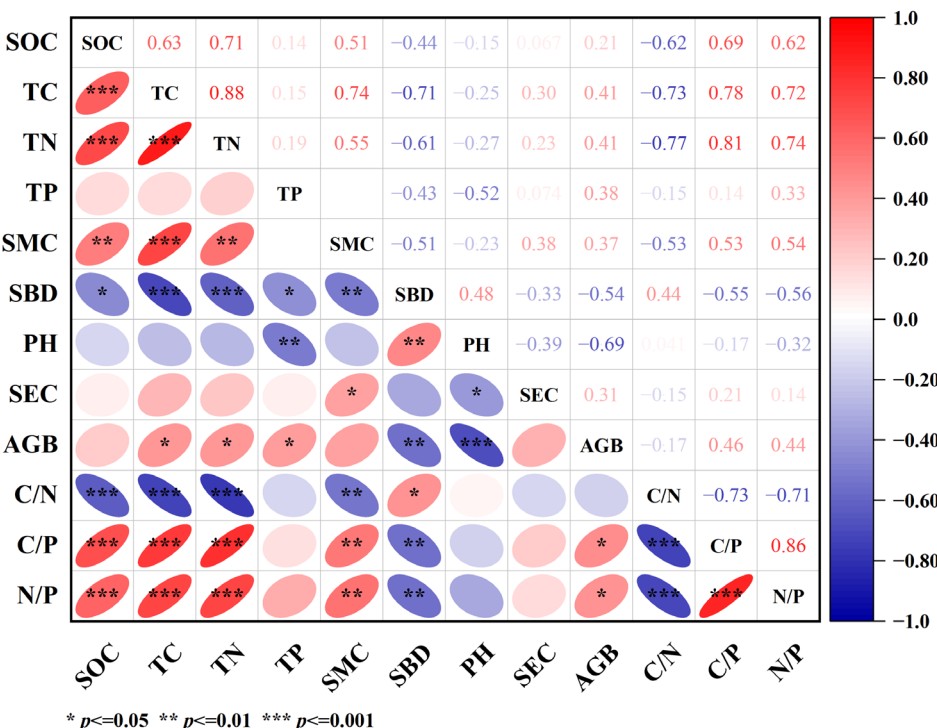

**Figure 7.** Correlation of soil properties with soil C, N and P contents and their biogenic stoichiometric characteristics.

## 4. Discussion

### 4.1. Effects of Spartina alterniflora Invasion on Soil C, N and P Contents

The present study showed that the SOC content of MF was significantly lower than that of soil after *Spartina alterniflora* invasion, which is similar to the findings of Yang, W et al. [19,50,51]. Our research reveals that the invasion of *Spartina alterniflora* does not increase or decrease the soil SOC content uninterruptedly, but decreases when a certain level is reached, unlike the results of Huang, X [30]. MF soil SOC content was evenly distributed at soil depth, and the vertical distribution pattern of soil SOC was significantly altered by the invasion of *Spartina alterniflora.* In this study, the increase in surface soil SOC was more significant than the increase in deep soil SOC after the invasion of *Spartina alterniflora* [45]. In general, SOC in coastal wetland soils is mainly derived from plant and animal debris, excreta and plankton and benthic organisms in seawater [52]. The study area has little human activity and is therefore less affected by pesticides, *Spartina alterniflora*, etc. Mutual rice grass has a sturdy root system and well-developed stem and leaf tissue, which gives it a high aboveground biomass and a well-developed root system. Some scholars found that the invasion of *Spartina alterniflora* increased primary productivity and decomposition rates of apoplastic matter compared to native species, thus increasing soil SOC stocks [24,53,54]. The well-developed root system and stem and leaf tissues of the *Spartina alterniflora* also weaken the tidal power and retain the organic matter of terrestrial and marine origin entrained by the river runoff at high and low tides for continuous deposition [25].

*Spartina alterniflora* invasion affects the N cycle in coastal wetlands along with the C cycle [20,32,55]. In the present study, there is a strong correlation between soil TN content and soil SOC content, and the trends in content were very similar across invasion stages and vertical distribution patterns, which is consistent with the results of studies at the Yangtze estuary and other sites [54]. The invasion of *Spartina alterniflora* promotes an increase in the species and numbers of nitrogen-fixing bacteria in the soil, which in turn converts the N in the soil into a more plant-friendly form for uptake [56]. In addition, more than 50% of the roots of the plant are concentrated in the topsoil layer, so the nutrients returned by

the decomposition of the roots of the plant are also concentrated in the topsoil layer. In addition, while the plants of *Spartina alterniflora* trap organic matter from seawater, they also trap N brought in by the tide, allowing N to accumulate on the soil surface. The present study found that *Spartina alterniflora* invasion did not significantly increase soil TP content, which is inconsistent with Wang's findings [57], probably because P cycling processes are more conservative than those of C and N [58]. The sources of P are complex and include both organic and inorganic plant debris, so we should further analyze TP in more detail in the future.

Understanding the invasion dynamics of *Spartina alterniflora* is helpful to intrusion management. Research shows that SOC and TN increase gradually with the extension of intrusion time [59]. In this study, the invasion of *Spartina alterniflora* increased the content of C and N in soil, but this increase was not continuous. After 16 years of invasion, the content of C and N began to decline. This is very similar to the research results of Xu et al., who showed that the maximum soil SOC content after the invasion of *Spartina alterniflora* appeared 18 years after the invasion, and then began to decline [43]. Yang et al. also showed that the soil N cycling rate decreased 20 years after the invasion of *Spartina alterniflora* [60]. The C and N in the soil of *Spartina alterniflora* mainly come from plant residues, roots and litter. With the increase in invasion years, the amount of these organic substances increases, and the soil accumulates C and N. However, with the increase in invasion years, the biomass of *Spartina alterniflora* decreased, making the content of C and N in soil decrease. The influence of the invasion time of *Spartina alterniflora* on P content is mainly divided into the consumption stage of P content in the initial growth stage of *Spartina alterniflora*, the transformation stage of inorganic P into organic P in the peak stage, and the decline stage of P content in the decline stage [39]. This may explain the phenomenon that the content of TP in soil temporarily increased and then decreased after the invasion of *Spartina alterniflora* in this study

*4.2. Effects of Spartina alterniflora Invasion on Soil Carbon, Nitrogen and Phosphorus Ecostoichiometry*

Ecostoichiometry, as an important indicator, can evaluate soil quality. However, there are few studies on the stoichiometric characteristics of C, N and P caused by the invasion of *Spartina alterniflora*, and most of the existing studies have the problem of large error due to different sampling depths. In this study, soil C/N after *Spartina alterniflora* invasion was significantly lower than that in MF, which was similar to the results of Liu et al. [61]. We found that the soil C/N value of *Spartina alterniflora* wetland in Dafeng Elk Nature Reserve of Yancheng is between 25–35, while in the study of Jin et al., the soil C/N value of *Spartina alterniflora* wetland in the Minjiang estuary is about 11 [62]. This difference shows that the influence of *Spartina alterniflora* invasion on soil C/N is very different in different regions. The level of soil C/N can indicate the transformation of humus in soil, the mineralization of soil N and the source of SOC. We found that soil C/N decreased and then increased with the increase in the invasion years of *Spartina alterniflora*, indicating that the invasion of *Spartina alterniflora* alters the proportion of terrestrial and marine soil organic carbon. This may be due to the fact that *Spartina alterniflora* has the function of promoting siltation. At the same time, with the increase in invasion time, its plants and roots change the source of C, thus changing the source of C [63].

C/P is a reflection of the mineralization ability of the soil P element, which can indicate the availability of P in soil. In this study, the average value of soil C/P is 15.913, far lower than the national average level of wetlands [64]. This shows that the microbial activity in our study area is less limited by P, and the soil P has higher biological activity [65]. We found that the soil C/P ratio continued to increase in the early period of *Spartina alterniflora* invasion (0–16 years), because the vegetation had a low demand for P during the early invasion. With the growth of the vegetation, the demand for P increased, so C/P increased. N/P in soil can be used to identify the nutrient supply status of soil after *Spartina alterniflora* invades wetlands. In this study, the soil N/P increased significantly after the invasion

of *Spartina alterniflora*, indicating that the soil fertility became better after the invasion of *Spartina alterniflora*. With the increase in invasion years, N/P increased (0–16 years), which indicates that the growth of *Spartina alterniflora* was severely restricted by N. At the same time, the increase in P required by plant growth also led to the decrease in soil N/P (16–26 years) [66].

*4.3. Correlation of Soil Properties with Soil C, N and P Content*

The invasion of *Spartina alterniflora* changed the physicochemical characteristics of the soil, such as pH, SBD, SMC and SEC, which indirectly led to changes in soil SOC, TN and TP contents. SBD has a strong correlation with soil SOC. The change in SBD affects the growth of plant roots and the activity of microorganisms in soil, thus affecting the TN and SOC of soil [67]. We found that there was obvious negative feedback regulation between SOC and SBD (Figure 7), in accordance with the research conclusions of others [68]. The invasion of *Spartina alterniflora* also increases the SMC of the soil. The change in SMC affects the soil permeability, changes the mineralization and decomposition rate of organic matter in the soil, and leads to the change in the content of SOC and TN [69]. We also found that SOC also interacted with TN and TP, and the increase in nitrogen and phosphorus enhances the productivity of plants, so as to accumulate more SOC, which in turn promotes the release of TN and TP from the soil by the decomposition process [70].

**5. Conclusions**

In the study, we measured and calculated soil physicochemical properties and SOC, TN and TP contents, stocks and stoichiometric ratios in the Dafeng Elk Nature Reserve under the invasion of mutual rice grass through field investigations and laboratory analyses, and came to the following conclusions:

(1) The storage of soil SOC and TN accumulates year by year with the increase in the invasion years of *Spartina alterniflora*, but their content stops increasing after a period of invasion and shows a downward trend. The invasion of *Spartina alterniflora* had little effect on soil TP content, which only increased at the beginning of the invasion and then decreased.

(2) *Spartina alterniflora* invasion changed soil properties, thus indirectly affected the accumulation of soil C, N and P. The changes in SOC, TN and TP were positively correlated with SEC and SMC, and negatively correlated with SBD and pH.

(3) After *Spartina alterniflora* invaded, the C/P ratio and N/P ratio increased significantly, while the C/N ratio decreased.

Changes in soil C, N and P contents and stocks indicate that the invasion of *Spartina alterniflora* has greatly changed the ecosystem processes and functions of coastal wetlands in eastern China. Future research is needed to document the dynamic changes in the nutrients and soil physicochemical properties of the plants themselves during the invasion of *Spartina alterniflora*, in order to provide theoretical support for the future management of coastal wetlands.

**Author Contributions:** Conceptualization, Y.S., D.Y. and Z.L.; methodology, Y.S. and D.Y.; formal analysis, Y.S. and J.L.; investigation, Y.S., D.Y. and Z.L.; resources, Z.L.; data curation, Y.S., S.X., Y.L., L.C., M.L. and C.W.; writing—original draft preparation, Y.S.; writing—review and editing, Y.S., D.Y. and Z.L.; supervision, D.Y. and Z.L.; project administration, D.Y. and Z.L. All authors have read and agreed to the published version of the manuscript.

**Funding:** This research was funded by the National Natural Science Foundation of China (No. 41871097) and the Priority Academic Program Development of Jiangsu Higher Education Institution (PAPD).

**Data Availability Statement:** Not applicable.

**Acknowledgments:** The authors gratefully acknowledge editors and reviewers for raising suggestions and comments on this paper.

**Conflicts of Interest:** The authors declare no conflict of interest.

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
