# Peer review of "Effects of Spartina alterniflora Invasion on Soil Carbon, Nitrogen and Phosphorus in Yancheng Coastal Wetlands"

_land, doi:10.3390/land11122218_

Round 1

Reviewer 1 Report

This article is hardly innovative. Similar to a large number of studies on Spartina alterniflora, even listed clearly in the preface literature review. Attempt to proceed from the perspective of stoichiometric ratio, but no corresponding analysis was carried out.

Since the invasion duration is an innovation extracted, the discussion part should analyze the time effect of invasion, but it does not.

Soil C, N and P dynamics should be most directly related to Spartina alterniflora biomass or other related plant indicators. Your data seems to show that, too. But instead of focusing on the plants, you focus on the indirect effects of the physical properties of the soil.

The article was sloppy in many details. For example, many unusual abbreviations are not marked with their full names. The line chart has no error bar. The superscript, subscript and upper and lower case letters in the chart are inconsistent.

In the last paragraph of the discussion section, since there is no measurement of the variables you mentioned, do not arbitrarily include them in your study to discuss.

Reviewer 2 Report

Abstract

General: Abstract is weakly developed. Reflection of more summarised results is desirable.

Line 27: Hibiscus sabdariffa or Spartina alterniflora?

Introduction

Line 34: Please correct “I It has…”.

Line 51: Please, provide an abbreviation the first time a word is mentioned. For instance, “carbon”, “nitrogen” and “phosphorus” is mentioned also above (line 37 and 38).

Line 54: If an abbreviation is given once, please use the abbreviation in the following text. For instance, in lines 54-56, lines 73-74, line 77, line 83, line 96, lines 103-106 and other.

 Materials and Methods

Line 131: Please format latin names (for instance, Spartina alterniflora in line 131, 133, 135, 139 and other) in italic through the text.

Lines 154-156: Please provide full names/explanations of abbreviations “SBD”, “SWC” and “SEC”.

Line 156: Please correct “weremeasured”.

 Results

Table 1. I suggest replace “g/m2” and “n/m2” with “g m-2” and “n m-2”. Please use this type of units here and in the following text including figures.

Table 1. What in shown after "±" (mean ± S.E.)? Please add explanation.

Figure 2: Is the mudflats soil used as a control against which to compare the impacts of Spartina alterniflora invasion? If yes, explain it in the methods and materials section.

Line 224-236: Both “soil total nitrogen” and “total soil nitrogen” is used. Please use consistently.

Line 254: Please correct “significant.he”.

 Discussion

General: Discussion part is weakly developed. Missing discussion of soil C, N, P stoichiometric ratios.

Line 356: Please correct “affecting t litter”.

Figure 7: I suggest to transfer Figure 7 from discussion part to results section.

Conclusion

General: Please do not repeat the results, but try to draw conclusions about the main trends/patterns.

Round 2

Reviewer 1 Report

The article has certainly been revised and updated a lot, but I think there is still room for improvement in the discussion section. It can be accepted after improving your writing mentality. 

4.2 For the discussion of stoichiometric ratio, what we want to see is the differences and changes in the environmental ecological significance reflected behind the ratio and the reasons for them, just as you mentioned in lines 57-59 and 368. It is not simply explained that the different contents of certain elements lead to different stoichiometric ratios, which is a very simple truth. It is not just to compare with the reported results of others, and to show that there are differences between regions, which is not the focus of this article.

Reviewer 2 Report

Thank you for your work, the manuscript is significantly improved.

Only some minor improvements are still necessary to be implemented:

Line 288: Please correct "char-acteristics".

Line 323:  Please use space in "[57],probably". Please check space usage through the text.

Line 324:  Please use space in "N[58].The". Please check space usage through the text.

Line 329: Please correct "content of C N in soil,".

Line 350-351: Meaning of this sentence "Changes in soil TC and TN levels due to Spartina alterniflora invasion" is not clear.

Line 364: Please correct "T This is because...".
